# Competition of Plants and Microorganisms for Added Nitrogen in Different Fertilizer Forms in a Semi-Arid Climate

Markus Koch [1,2,*], Kanat Akshalov [3], Jannis Florian Carstens [1,4], Olga Shibistova [1,5], Claus Florian Stange [6], Simon Thiedau [1], Alfiya Kassymova [7], Leopold Sauheitl [1], Tobias Meinel [8] and Georg Guggenberger [1]

1    Institute of Soil Science, Leibniz University Hanover, Herrenhäuser Str. 2, 30419 Hanover, Germany; carstens@ifbk.uni-hannover.de (J.F.C.); olgas@ifbk.uni-hannover.de (O.S.); simonthiedau@gmail.com (S.T.); sauheitl@ifbk.uni-hannover.de (L.S.); guggenberger@ifbk.uni-hannover.de (G.G.)
2    Department of Geoecology, Martin Luther University Halle-Wittenberg, von-Seckendorff-Platz 4, 06120 Halle, Germany
3    Scientific and Production Center for Grain Farming Named after A. I. Baraev, Baraev Str. 15, Shortandy 021601, Kazakhstan; kanatakshalov@mail.ru
4    Technische Universität Clausthal, Adolph-Roemer-Straße 2A, 38678 Clausthal-Zellerfeld, Germany
5    VN Sukachev Institute of Forest, Siberian Branch of the Russian Academy of Sciences, 660036 Krasnoyarsk, Russia
6    Federal Institute for Geosciences and Natural Resources (BGR), Stilleweg 2, 30655 Hannover, Germany; Florian.Stange@bgr.de
7    JSC Atameken-Agro, Estern Industrial Zone Driveway 20, Building 30, Kokschetau 020000, Kazakhstan; kasymova.alfiya97@mail.ru
8    Amazonen-Werke H. Dryer GmbH & Co. KG, Am Amazonenwerk 9-13, 49205 Hasbergen, Germany; Dr.Tobias.Meinel@amazone.de
*    Correspondence: koch@ifbk.uni-hannover.de

**Abstract:** In nitrogen (N) -limited agricultural systems, a high microbial immobilization of applied fertilizer-N can limit its availability to plants. However, there is scarce information on the effect of the form of fertilizer used on the plant–microorganism competition in clay-rich soils under a severe semi-arid climate. In a field study, we investigated the wheat–microorganism competition after the direct application of $NH_4{}^{15}NO_3$ closely to seeds in arable fields in North Kazakhstan, documenting the effect of the use of liquid versus granular fertilizer and mini-tillage versus no-tillage. Our results barely showed any fertilizer-N translocation in the soil. Plants outcompete microorganisms for fertilizer-N during the vegetation period. Microbial-to-plant $^{15}N$ ratios revealed a predominant fertilizer-$^{15}N$ uptake by plants. The strong competition for N was mainly related to the placement of the fertilizer close to the seeds. Moreover, the long time interval between fertilization and sampling enhanced the competition for N, meaning that previously microbially immobilized N became available to plants through the death of microorganisms and their subsequent mineralization. The fertilizer distribution between microorganisms and plants did depend on the form of fertilizer used, owing to the good solubility of granular fertilizer. The smaller fertilizer-N uptake under the no-tilling condition was probably due to the more intense soil compaction, which caused a reduction in plant growth. The application of fertilizer close to the seeds and the small fertilizer translocation during the vegetation period ultimately resulted in a high level of plant N being derived from the fertilizer.

**Keywords:** liquid fertilizer; granular fertilizer; mini-till; no-till; ammonium nitrate; $^{15}N$

## 1. Introduction

Due to its co-variation, in semi-arid regions, besides water, nitrogen (N) is a major factor limiting the productivity and quality of wheat [1], as well as the growth and metabolism of microorganisms [2,3]. Consequently, plants and microorganisms may strongly compete against each other for available N [4]. Microbial N immobilization occurs when the C:N

ratio of the decomposed substrate is higher than that of microorganisms (after taking the already microbially respired $CO_2$ into account) [5]. Due to their larger surface area-to-volume ratio and rapid growth, microorganisms have been assumed to out-compete plants for N [5]. In annual grasslands, microorganisms can assimilate nitrate [6] and ammonium [7] two and nine times faster than plants within the first 24 h after N application. Microorganisms may directly assimilate fertilizer N after wetting in spring [8,9], whereas crop plants acquire N mainly in the vegetative and reproductive growth stages [10]. Hence, combined seeding and fertilization may be problematic for the efficient N use of plants.

The competition of plants and microorganisms for N is mostly tested by the application of $^{15}$N-labeled ammonium, nitrate, or amino acids to the soil N pool and then measuring the $^{15}$N in plants and organic and inorganic N forms a few hours to days (short-term) or weeks to months (long-term) later [11]. In $^{15}$N studies on temperate humid and Mediterranean grassland soils, microorganisms were found to be strong competitors and reported to assimilate >60% of added N [5,12,13]. In a field experiment carried out in tallgrass prairie, up to 46% of added $^{15}$N was found to be immobilized by microorganisms [14]. However, after the rapid initial N capture of microorganisms, C limitation causes a halt in microbial growth and hence N acquisition [5]. Several days to months after the N application, in situ and in vitro studies in humid and temperate grassland soils showed that most (45% to 96%) of the added $^{15}$N was recovered in plants, whereas only 0% to 15% was recovered in microbial biomass [5,11,12,15]. Hence, in the long run plants out-competed microorganisms, as shown in semi-arid prairie (USA) and steppe soils (Inner Mongolia) [5,16,17]. Due to microbial death and reassimilation, microbial assimilated N may several times contribute to the soil N pool. Consequently, formerly microbially assimilated N reenters the plant–microorganism competition, whereas plants store captured N over longer time periods [5,11]. Hence, in the long run plant–microbe N competition is the result of numerous short-term competitions [11] and mainly governed by the residence time of N in both pools [17].

Though the short- and long-term competition of plants and microorganisms has been well investigated in temperate and humid climatic conditions, there is a gap in our knowledge concerning how the application of different fertilizer forms and land use management types may affect the plant–microorganism competition in a semi-arid climate. In general, liquid N fertilizer can enhance nutrient uptake and yield as compared to its granular counterparts, as shown in Mediterranean Australia [18,19]. This is because of the immediate availability of nutrients and higher diffusion occurring in the soil when applied in liquid form, as shown under more humid conditions [18,20]. The widely adapted conservational tillage (mini-till and no-till) used in Kazakhstan improved biological and physical parameters in the soil compared to conventional tillage [21]. No-till soil management does not include any soil management until after the last harvest. At sowing time, seeds are directly seeded. Under mini-till management, in contrast, the top soil layer is shallowly managed up to a few cm by a cultivator before sowing is completed. For both conservational tillage forms, favorable conditions for microbial growth have been found [21]. However, it is unknown to what extend the presumed better N availability of liquid fertilizer impacts the plant–microorganism competition for N in different conservational tilled croplands. Our objective was thus to test the plant–microorganism competition for N supplied in liquid and granular form during seeding carried out under field conditions using the tillage forms of no-till and mini-till in semi-arid North Kazakhstan. North Kazakhstan was chosen as our study region as it represents a global bread basket [22,23]. However, there have only been a few studies on the N turnover in this huge area to date. Soils in this region were formerly subjected to unsustainable land use for many years, which resulted in soil degradation [24–26] and low contents of mineralized N [27]. We hypothesized that (i) the use of liquid fertilizer can increase plant growth and N uptake in these semi-arid regions, regardless of the tillage form used, thus increasing plant competition for N in the long run. Furthermore, we assume that (ii) the plant–microorganism competition occurring in semi-arid, clay-rich soils is more severe than that reported for more humid regions.

## 2. Materials and Methods

### 2.1. Study Site and Basic Soil Characteristics

In 2019, a $^{15}N$ labeling experiment was conducted on the site of the Scientific and Production Center for Grain Farming, which is named after A. I. Baraev, Shortandy, Akmola District, Northern Kazakhstan. The area belongs to the semi-arid steppe zone, with a mean annual temperature and mean annual precipitation of 1.8 °C and 324 mm (Baraev Institute), respectively. During the 2019 vegetation period, from the end of May to the beginning of September the mean daily temperature was 16.4 °C and the total cumulative precipitation amounted to 92 mm (Figure 1). For both fields, similar basic soil parameters were observed, without any significant differences being noted in any soil parameter for each soil increment (Table 1). Soils were shown to be fine textured (silt loam) throughout the soil profile, with high contents of clay and silt. The no-till field showed 4% higher total C and 6% higher total N contents on average than the mini-till field. However, in the no-till field the surface soil (0–20 cm) was more compacted (1.3 g cm$^{-3}$) than in the mini-till field (1.1 g cm$^{-3}$). Soils are characterized as Southern Chernozems according to the local soil classification and as Typic Haplustolls after US Taxonomy [28,29].

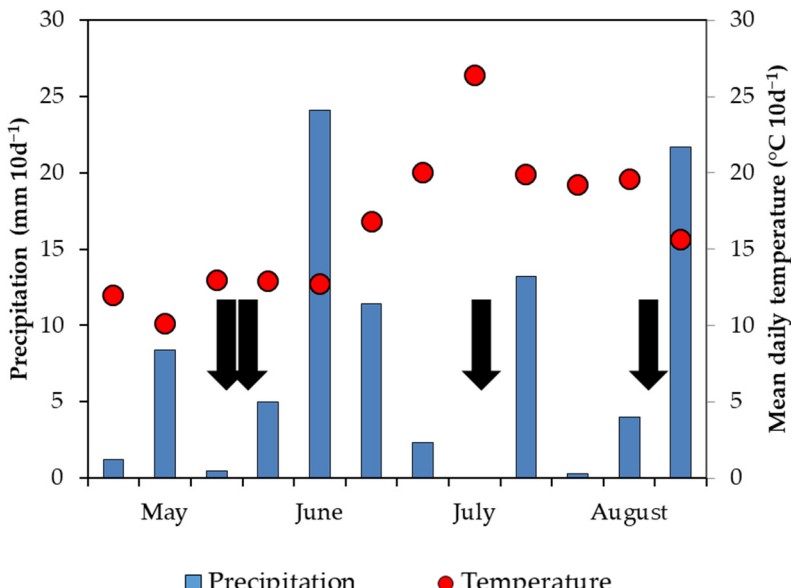

**Figure 1.** Climate data for the vegetation period 2019 in Shortandy, Kazakhstan. The cumulative precipitation and mean daily temperature are given in 10-day intervals. Samplings are indicated by vertical black arrows.

**Table 1.** Basic soil parameter for no-till and mini-till management for the soil depths of 0–20, 20–40, and 40–60 cm. Values were determined before the start of the experiment in May 2019. Texture was determined by pipette analysis [30]. The means of n = 2 samples ± the standard deviation are given.

| Tillage | Depth | Clay | Silt | Sand | Total C | Total N | C/N | Bulk Density |
|---|---|---|---|---|---|---|---|---|
| | (cm) | | | (g kg$^{-1}$) | | | (-) | (g cm$^{-3}$) |
| Mini-till | 0–20 | 369 ± 17 | 603 ± 11 | 28 ± 6 | 29.1 ± 0.1 | 2.2 ± 0.0 | 13.2 ± 0.1 | 1.1 ± 0.1 |
| | 20–40 | 419 ± 22 | 552 ± 13 | 29 ± 8 | 27.5 ± 0.3 | 1.9 ± 0.0 | 14.7 ± 0.1 | 1.2 ± 0.0 |
| | 40–60 | 420 ± 29 | 556 ± 27 | 24 ± 3 | 26.4 ± 0.5 | 1.5 ± 0.1 | 17.2 ± 0.7 | 1.3 ± 0.0 |
| No-till | 0–20 | 406 ± 12 | 570 ± 16 | 24 ± 4 | 29.9 ± 0.3 | 2.3 ± 0.0 | 12.8 ± 0.0 | 1.3 ± 0.1 |
| | 20–40 | 395 ± 19 | 583 ± 18 | 22 ± 2 | 29.9 ± 1.6 | 2.1 ± 0.1 | 14.5 ± 0.3 | 1.1 ± 0.1 |
| | 40–60 | 404 ± 2 | 567 ± 1 | 29 ± 3 | 26.8 ± 0.4 | 1.5 ± 0.0 | 17.6 ± 0.2 | 1.2 ± 0.0 |

## 2.2. Experimental Setup and Sampling

Two adjacent arable fields were chosen for the $^{15}$N labeling experiment. One field was managed under mini-till (51°35.585′ N, 071°03.624′ E), while the other field was managed under no-till and direct seeding (51°35.615′ N, 071°03.707′ E) for at least 20 years. Shortly before seeding, the mini-till field was mechanically tilled to a 5–6 cm depth, while at the same time the no-till field was treated with glyphosate. In the mini-till field, 1.95 t ha$^{-1}$ of straw remained in the field after the last harvest, while in the no-till field the amount was 2.15 t ha$^{-1}$. The C/N in shoots in the mini-till field was significantly higher than that in the no-till field (Table 2). Both fields were simultaneously seeded and fertilized with granular ammonium nitrate at a rate of 20 kg N ha$^{-1}$. Both fields were in a wheat–wheat–fallow crop rotation, with the last fallow period occurring in 2017.

**Table 2.** Carbon-to-nitrogen ratio in plant compartments of the leaf and shoot, as determined at the last sampling time point in August. The means of n = 3 samples $\pm$ the standard deviation are given. Superscripted letters indicate statistically significant differences between treatments (tillage form $\times$ fertilizer form).

| Tillage Form | Fertilizer Form | Compartment | C/N |
|---|---|---|---|
| Mini-till | Liquid | Leaf | 46.9 $\pm$ 8.1 [a] |
| | | Shoot | 120.6 $\pm$ 13.7 [a] |
| Mini-till | Granular | Leaf | 51.8 $\pm$ 9.5 [a] |
| | | Shoot | 116.9 $\pm$ 24.3 [a] |
| No-till | Liquid | Leaf | 42.1 $\pm$ 2.3 [a] |
| | | Shoot | 82.5 $\pm$ 12.8 [a] |
| No-till | Granular | Leaf | 45.3 $\pm$ 3.3 [a] |
| | | Shoot | 94.4 $\pm$ 18.5 [a] |

On 30 May 2019, summer wheat (*Triticum aestivum* subsp. *Aestivum*) was directly seeded by a drill cultivar (Condor 1201 C, Amazone, Hasbergen, Germany) at a seeding rate of 120 kg ha$^{-1}$ with a furrow distance of 25 cm. In total, we tested two treatment pairs (granular versus liquid fertilization, mini-till versus no-till) and their interactions. Within each of the two fields with different tillage conditions, 8 subplots with the size of 75 $\times$ 130 cm were established randomly. Of these, six subplots were used to test the effectiveness of the fertilizer form used (liquid versus granular) in triplicate, while two subplots served as controls. All subplots were trenched to the depth of 30 cm and isolated with plastic foil. This was carried out to avoid (i) fertilizer-N uptake by neighboring plants, (ii) horizontal leaching, and (iii) the surface runoff of fertilizer-N. The cut-off zones of these trenches were refilled with soil and re-seeded by hand outside of each plot.

Ammonium nitrate fertilizer (NH$_4$$^{15}$NO$_3$, 16.7 at% excess $^{15}$N) was prepared by mixing 98 at% NH$_4$$^{15}$NO$_3$ (Campro Scientific, Berlin, Germany) with commercial unlabeled NH$_4$NO$_3$ (calculated after [31]), which was applied to the soil on the day of seeding. To apply the fertilizer in liquid form, the NH$_4$$^{15}$NO$_3$ was dissolved in distilled water prior to fertilization. Granular fertilizer was produced by pressing the NH$_4$$^{15}$NO$_3$ at 100 bar for 9 min in a FTIR IR press (Beckman, 00-25, Glenrothes, UK). Labeled fertilizer was applied to each subplot by hand in the furrow (3 furrows per subplot) at a rate of 20.5 kg N ha$^{-1}$ with a seeding depth of 3 cm directly next to the seeds. Fertilizer application should, in both cases, mimic realistic fertilizer application in Kazakhstan. For the liquid fertilization, 0.1 mL of dissolved NH$_4$$^{15}$NO$_3$ (5.8 M) was applied every 3 cm using a Ripette® (Ritter, Schwabmünchen, Germany). For granular fertilizer, fertilizer tablets were applied in the furrow at distances of 20 cm. For this, tablets were crushed and mixed with the surrounding soil over a length of 5 cm in each direction within the furrow (we made sure that seeds were not relocated). Hence, the actual distance between the fertilizer granules was about 10 cm, but fertilizer hotspots might have developed. Granular fertilizer was applied in this way because, over the past few years, we had usually observed the fertilization of granular fertilizer along the furrows in Kazakhstan and no continuous bands of fertilizer. Control

subplots received unlabeled commercial ammonium nitrate fertilizer (GOST 2-2013) in liquid and granular form.

### 2.3. Soil and Plant Sampling

Soils from all subplots were sampled four times during the vegetation period. The first two samplings were carried out two days before and two days after seeding and fertilization. In these sampling points, the soils of each subplot were sampled at three locations directly under the furrow with cylindrical cores. Therefore, in each sampling spot the soil was dug until a 60 cm depth was reached. In each 0–20, 20–40, and 40–60 cm soil depth increment, three cylindrical cores were taken and each homogenized to a mixed sample. For the 0–20 cm soil depth increment, the first 0–4 cm of topsoil was not included in the soil sampling in order to avoid the direct sampling of fertilizer. Plant residues were removed from the soil, and samples were oven-dried at 40 °C.

The third and fourth soil samplings were conducted in July and August during the steam extension stage and the ripening stage of plant growth. These sampling time points were selected as nitrogen was highly allocated within the plants at these vegetation stages [10,32]. To account for the spatial heterogeneity of fertilizer distribution in the soil, subplots were sampled from three spatially distributed $20 \times 25$ cm microplots within each subplot. In these microplots, soil samples were taken with an $N_{min}$ corer on two randomly chosen $5 \times 5$ cm squares, meaning that furrows, hills, and their interspace were sampled. Both these samples were homogenized in a mixed sample (a total of nine soil samples were generated from each subplot). Plant residues were removed and soil samples were oven-dried at 40 °C.

Aboveground wheat plant biomass growing in the furrow was collected from each microplot by cutting off the plants directly at the soil surface. Fresh plants were then separated into shoots, leaves, and grains. Each plant compartment was weighted in its fresh and 40 °C oven-dried state to determine the plant biomass.

### 2.4. $\delta^{15}N$ Analysis

Nitrate was extracted from fresh soil samples within 30 h after sampling in a 12.5 mM $CaCl_2$ solution with a soil-to-volume ratio of 1:4 (*w:v*) [33]. Extracts were shaken for 1.5 h and filtered <0.45 μm using cellulose acetate syringe filters (Berrytech, Grünwald, Germany). All extracts were poisoned with HgCl (350 mg $L^{-1}$) to prevent microbial N transformation. Extracts were stored cold in the dark until their content of nitrate and its $^{15}N$ abundance were determined in $VCl_3$ using the SPINMas technique [34]. Nitrate was our main focus because of the rapid oxidation of ammonium to nitrate. Ammonia was not measured because, over our last few years of working in Kazakhstan, we always found very minor ammonium contents in these clayey and dry soils. It was also reported that the $NH_4$-N contents in the soils of Northern Kazakhstan are usually small and do not exceed 3 to 4 mg $kg^{-1}$ [27,35].

Unfortunately, it was not possible to directly measure microbial biomass N and $^{15}N$ (MBN) in fresh samples; thus, these were determined in air-dried soil samples [36–39]. The microbial biomass N in soil samples from a 0–20 cm depth was analyzed using a modified chloroform–fumigation–extraction (CFE) method, which includes pre-extraction with 0.05 M $K_2SO_4$ prior to the CFE procedure [37,40]. Microbial biomass N was determined at a 0–20 cm depth in order to investigate the microbial fertilizer immobilization directly at the application spot. Pre-extraction should remove high amounts of mineralized N forms to enhance the determination of possibly small microbial $^{15}N$. About 30 g of each dried sample was rewetted to a 60% water holding capacity and incubated for 2 weeks at room temperature. Afterwards, the samples were sieved <2 mm and the water contents were determined. Each sample was divided into two equal parts. Both aliquots were pre-extracted with 0.05 M $K_2SO_4$ (1:4 *w:v*) for 30 min by horizontal shaking at 200 rev. $min^{-1}$. Afterwards, one sample was fumigated to cause the lysis of microbial cells; the other sample was not fumigated. Fumigation was conducted under vacuum (800 mbar) for

24 h at 25 °C with ethanol-free chloroform in a desiccator. After chloroform was removed, the fumigated and unfumigated samples were extracted with 0.5 M $K_2SO_4$. Extracts were then filtered (Satorius hw3, Göttingen, Germany) before measurement to determine their total organic carbon (TOC) and total nitrogen (TN) contents on a multiN/C 2100S automatic analyzer (Analytik Jena AG, Jena, Germany). Afterwards, the extracts were freeze-dried and measured on an EA-IRMS (vario ISOTOPE elemental analyzer coupled with an isoprime precisION isotope ratio mass spectrometer, Elementar Analysensysteme GmbH, Langenselbold, Germany) to determine their N contents as well as $\delta^{15}N$ ratios.

To determine the total $^{15}N$ abundance in soil and plants, all samples were dried at 40 °C. Soil samples were sieved <2 mm and visible plant material was removed. Plant samples were shredded. All samples were milled for isotope analysis on the EA-IRMS.

## 2.5. Data and Statistical Analysis

All data analyses were carried out in R 3.6.3. [41] and Excel 2010 (Microsoft). $\delta^{15}N$ ratios were transformed into at% as:

$$at\% = (100 * AR * (\delta^{15}N/1000 + 1))/(1 + (AR * (\delta^{15}N/1000 + 1))) \tag{1}$$

where at% is the atomic percentage of $^{15}N$, AR is the absolute ratio of mole fractions of $^{15}N$ in air of 0.0036764 [42], and $\delta^{15}N$ is the ratio of $^{15}N$ to $^{14}N$ in a sample to that of air as a standard.

Due to the spatially distributed soil sampling that took place in July and August, all soil-derived N-pools (based on mg kg$^{-1}$) and their $^{15}N$ values (based on at%) were interpolated by applying a linear model with a least square estimation, in which the sampling position (hill, furrow, and clearance) was used as a variable. Afterwards, the $^{15}N$ abundance in the samples were corrected for the background abundance and expressed as the at% excess.

Microbial biomass N was calculated from the difference between the fumigated and unfumigated TN. The $^{15}N$ enrichment in MBN was calculated from the mass balance [43].

The plant N uptake was evaluated as the plant dry weight (g per m$^2$) multiplied by its N content (%). The nitrogen harvest index (NHI) was calculated as the nitrogen content in the grain related to the N content of the whole plant, which are both given in g m$^2$:

$$NHI = N \text{ yield in grains}/N \text{ yield in plant} \tag{2}$$

The percentage of N derived from fertilizer (NdfF%) in plant samples was calculated as:

$$NdfF\ (\%) = {}^{15}N \text{ at\% excess in plant} * 100/{}^{15}N \text{ at\% excess in fertilizer} \tag{3}$$

The fertilizer N uptake by plants was calculated by the multiplication of the plant N yield with the NdfF. The $^{15}N$ recoveries in all soil and plant compartments were calculated as the % of applied $^{15}N$ (mg per subplot). To compare the $^{15}N$ recovery of the investigated N fractions at a given time, the $^{15}N$ recoveries in each compartment were additionally related to the total recovered $^{15}N$ at a given time point.

Plant–microorganism competition was calculated by two indexes. First, microbial biomass N-to-plant N ratios [44] were determined to assess the competition for N. The second competition index was calculated based as the ratio of $^{15}N$ recoveries in MBN to plant to determine the competition for fertilizer-N [17]. Hence, the first index describes the competition for N, whereas the second gives information about the competition for fertilizer N. For both calculations, the unit mg N per m$^2$ was used for microbial biomass and plants.

We used the aov-function in R to conduct two-way analysis of variance (ANOVA) to test for significant differences between fertilizer and tillage form and their interactions on aboveground dry weight plant biomass, plant N uptake, grain yield, MBN, NdfF, NHI,

as well as the $^{15}$N recovery of applied $^{15}$N in soil and plant, nitrate and MBN. If ANOVA assumptions were not met, data were log-transformed. Groups were compared using the Tukey's honest significant post hoc test (HSD) and statistical differences reported at a significance level of $p < 0.05$. Statistic significant different groups are presented as small letters in tables. Detailed ANOVA results are given in the Supplementary Table S3.

## 3. Results

### 3.1. Soil Nitrogen

Nitrate contents were small, at 0.4 to 1.9 g m$^{-2}$ (1.5 to 12.9 mg kg$^{-1}$) throughout the vegetation period (Table 3). In general, the highest but most greatly deviating nitrate concentrations were observed in June after the fertilizer application, when the soil was the wettest (Supplementary Table S1). These strong variations in nitrate contents indicate that the ammonium nitrate fertilizer from the top 0–4 cm was not yet well distributed into the sampled soil > 4 cm.

**Table 3.** Nitrogen contents in plant and soil (0–20 cm) compartments over the vegetation period. The means $\pm$ the standard deviation of n = 3 subplots are given. Abbreviations are: MBN: microbial biomass nitrogen. For empty cells, there was no plant material available at these time points. Superscripted letters indicate statistically significant differences between treatments (tillage form $\times$ fertilizer form) at a specific sampling date.

| Variant | Compartment | May | June | July | August |
|---|---|---|---|---|---|
| | | | (g m$^{-2}$) | | |
| Mini-till | NO$_3$-N | 1.2 $\pm$ 0.5 [a] | 2.5 $\pm$ 0.5 [a] | 0.8 $\pm$ 0.0 [a] | 0.9 $\pm$ 0.3 [a] |
| Liquid | MBN | 8.3 $\pm$ 1.6 [a] | 9.3 $\pm$ 1.7 [a] | 11.7 $\pm$ 0.5 [a] | 11.6 $\pm$ 1.3 [a] |
| | Plant | | | 5.9 $\pm$ 0.6 [a] | 10.7 $\pm$ 3.8 [a] |
| | Grain | | | | 8.6 $\pm$ 2.9 [a] |
| | Leaf | | | 4.4 $\pm$ 0.4 [a] | 1.2 $\pm$ 0.6 [a] |
| | Shoot | | | 1.5 $\pm$ 0.3 [a] | 0.9 $\pm$ 0.4 [a] |
| Mini-till | NO$_3$-N | 1.8 $\pm$ 0.6 [a] | 0.8 $\pm$ 0.4 [a] | 1.1 $\pm$ 0.0 [a, c] | 1.7 $\pm$ 0.1 [b] |
| Granular | MBN | 8.0 $\pm$ 2.0 [a] | 11.1 $\pm$ 1.2 [a] | 12.6 $\pm$ 0.3 [a,b] | 13.0 $\pm$ 1.6 [a] |
| | Plant | | | 5.8 $\pm$ 0.5 [a] | 9.9 $\pm$ 2.2 [a] |
| | Grain | | | | 7.9 $\pm$ 1.5 [a] |
| | Leaf | | | 4.2 $\pm$ 0.4 [a] | 1.1 $\pm$ 0.4 [a] |
| | Shoot | | | 1.6 $\pm$ 0.1 [a] | 0.9 $\pm$ 0.4 [a] |
| No-till | NO$_3$-N | 1.8 $\pm$ 0.3 [a] | 2.8 $\pm$ 1.5 [a] | 2.3 $\pm$ 0.4 [b] | 0.6 $\pm$ 0.1 [c,d] |
| Liquid | MBN | 9.2 $\pm$ 1.4 [a] | 11.1 $\pm$ 0.7 [a] | 13.7 $\pm$ 0.1 [a,b] | 11.9 $\pm$ 0.7 [a] |
| | Plant | | | 7.1 $\pm$ 0.6 [a] | 10.7 $\pm$ 2.3 [a] |
| | Grain | | | | 8.3 $\pm$ 1.7 [a] |
| | Leaf | | | 5.1 $\pm$ 0.6 [a] | 1.2 $\pm$ 0.2 [a] |
| | Shoot | | | 2.1 $\pm$ 0.0 [a] | 1.2 $\pm$ 0.4 [a] |
| No-till | NO$_3$-N | 1.9 $\pm$ 0.4 [a] | 1.9 $\pm$ 1.3 [a] | 1.5 $\pm$ 0.6 [a,b,c] | 0.4 $\pm$ 0.1 [d] |
| Granular | MBN | 9.8 $\pm$ 1.4 [a] | 10.6 $\pm$ 3.1 [a] | 16.0 $\pm$ 2.0 [b] | 12.1 $\pm$ 0.3 [a] |
| | Plant | | | 6.3 $\pm$ 1.8 [a] | 7.7 $\pm$ 1.3 [a] |
| | Grain | | | | 5.9 $\pm$ 1.3 [a] |
| | Leaf | | | 4.4 $\pm$ 1.2 [a] | 1.0 $\pm$ 0.0 [a] |
| | Shoot | | | 1.9 $\pm$ 0.6 [a] | 0.8 $\pm$ 0.0 [a] |

Over the vegetation period, MBN ranged from 9.3 to 16.0 g m$^{-2}$ (41.9 to 63.5 mg kg$^{-1}$) (Table 3). For both tillage and fertilizer forms, MBN tended to increase about 7% to 27% from May to June (Table 3) and further in July. In July, MBN was significantly higher under granular fertilized treatments, while in August the MBN was significantly higher under the mini-till condition.

### 3.2. Plant Nitrogen Uptake

In July, wheat was similarly established under all treatments. In August, however, a shift in the vegetative stage of wheat was obvious, with less developed wheat plants

with milky grains and greener leaves seen under the no-till condition compared to already golden leaves and drier grains seen under the mini-till condition (about Zadoks 77 and 87, respectively) (Supplementary Figure S1a,b).

For all treatments, the above-ground plant biomass varied between 2.0 and 2.3 t ha$^{-1}$ in July and 5.4 and 7.3 t ha$^{-1}$ in August (Table 4), and grain yields ranged from 2.6 to 3.8 t ha$^{-1}$ (Table 4), in both cases without any significant differences being seen between plants treated with different fertilizer or tillage forms.

**Table 4.** Plant biomass in July and August and grain yield in August, shortly before harvest. the means $\pm$ the standard deviation of n = 3 subplots are given. Dry weight is abbreviated to d.w. Superscripted letters indicate statistically significant differences between treatments (tillage form $\times$ fertilizer form) at a specific sampling date.

| Tillage Form | Fertilizer Form | Total Plant d.w. (t ha$^{-1}$) July | Total Plant d.w. (t ha$^{-1}$) August | Grain Yield (t ha$^{-1}$) August |
|---|---|---|---|---|
| Mini-till | Liquid | 2.3 $\pm$ 0.1 [a] | 7.3 $\pm$ 2.3 [a] | 3.8 $\pm$ 1.1 [a] |
| Mini-till | Granular | 2.3 $\pm$ 0.4 [a] | 7.3 $\pm$ 1.1 [a] | 3.7 $\pm$ 0.4 [a] |
| No-till | Liquid | 2.1 $\pm$ 0.1 [a] | 6.2 $\pm$ 1.3 [a] | 3.2 $\pm$ 0.7 [a] |
| No-till | Granular | 2.0 $\pm$ 0.3 [a] | 5.4 $\pm$ 1.0 [a] | 2.6 $\pm$ 0.8 [a] |

At the first plant sampling in July, plants were already in the beginning of the jointing stage and thus were at a major N uptake stage [10], taking up 5.8 to 7.1 g N m$^{-2}$ (Table 3). Most plant N was stored in the leaves (Table 3) but was not affected by the fertilizer or tillage form used. In August, the plant biomass N ranged from 7.7 to 10.7 g m$^{-2}$ (Table 3) and was not significantly affected by the fertilizer or tillage form used. Most plant N was stored in the grains, and N storage in leaves and shoots decreased compared to the N contents in July (Table 3). Fertilizer form and tillage form did not affect the N contents in the plant compartments. The fertilizer-N uptake (NdfF) ranged from 1.0 to 1.6 g m$^{-2}$ (12% to 15% of the total N uptake; Table 5) and was significantly higher under the mini-till and liquid fertilization conditions than under the no-till and granular fertilization conditions. The NHI ranged from 0.77 to 0.81 and tended to be higher under the mini-till and liquid fertilization condition. However, these differences were not significant.

**Table 5.** Nitrogen derived from fertilizer (NdfF) in July and August and nitrogen harvest index (NHI) in August, shortly before harvest. the means $\pm$ the standard deviation of n = 3 subplots are given. Superscript letters denote differences between treatments (tillage form $\times$ fertilizer form) at a specific sampling date.

| Tillage Form | Fertilizer Form | NdfF (g m$^{-2}$) July | NdfF (g m$^{-2}$) August | NHI August |
|---|---|---|---|---|
| Mini-till | Liquid | 1.3 $\pm$ 0.1 [a] | 1.6 $\pm$ 0.1 [a,b] | 0.81 $\pm$ 0.03 [a] |
| Mini-till | Granular | 1.2 $\pm$ 0.2 [a] | 1.3 $\pm$ 0.1 [a,b,c] | 0.80 $\pm$ 0.03 [a] |
| No-till | Liquid | 1.1 $\pm$ 0.2 [a] | 1.5 $\pm$ 0.3 [a,b] | 0.79 $\pm$ 0.02 [a] |
| No-till | Granular | 1.3 $\pm$ 0.4 [a] | 1.0 $\pm$ 0.1 [c] | 0.77 $\pm$ 0.04 [a] |

### 3.3. $^{15}$N Recovery

In the beginning of June, three days after the fertilizer application, no fertilizer granules were found in the soil any more, as they has probably already been dissolved by the soil solution. In total, 18% to 33% of the applied $^{15}$N was recovered in the 4–60 cm soil (excluding the fertilization layer) (Figure 2), indicating the small translocation of fertilizer-N from the top 4 cm. However, about 8% $\pm$ 3% of the total recovered soil $^{15}$N was found in MBN, showing microbial immobilization of fertilizer-N as soon as the fertilizer-N became available to microorganisms. In total, 25% $\pm$ 16% of the total recovered soil $^{15}$N was found in nitrate. The high variability along with the small recovery indicates that most fertilizer

remained in the top 4 cm and was only slightly translocated. The differences in recoveries were not significant.

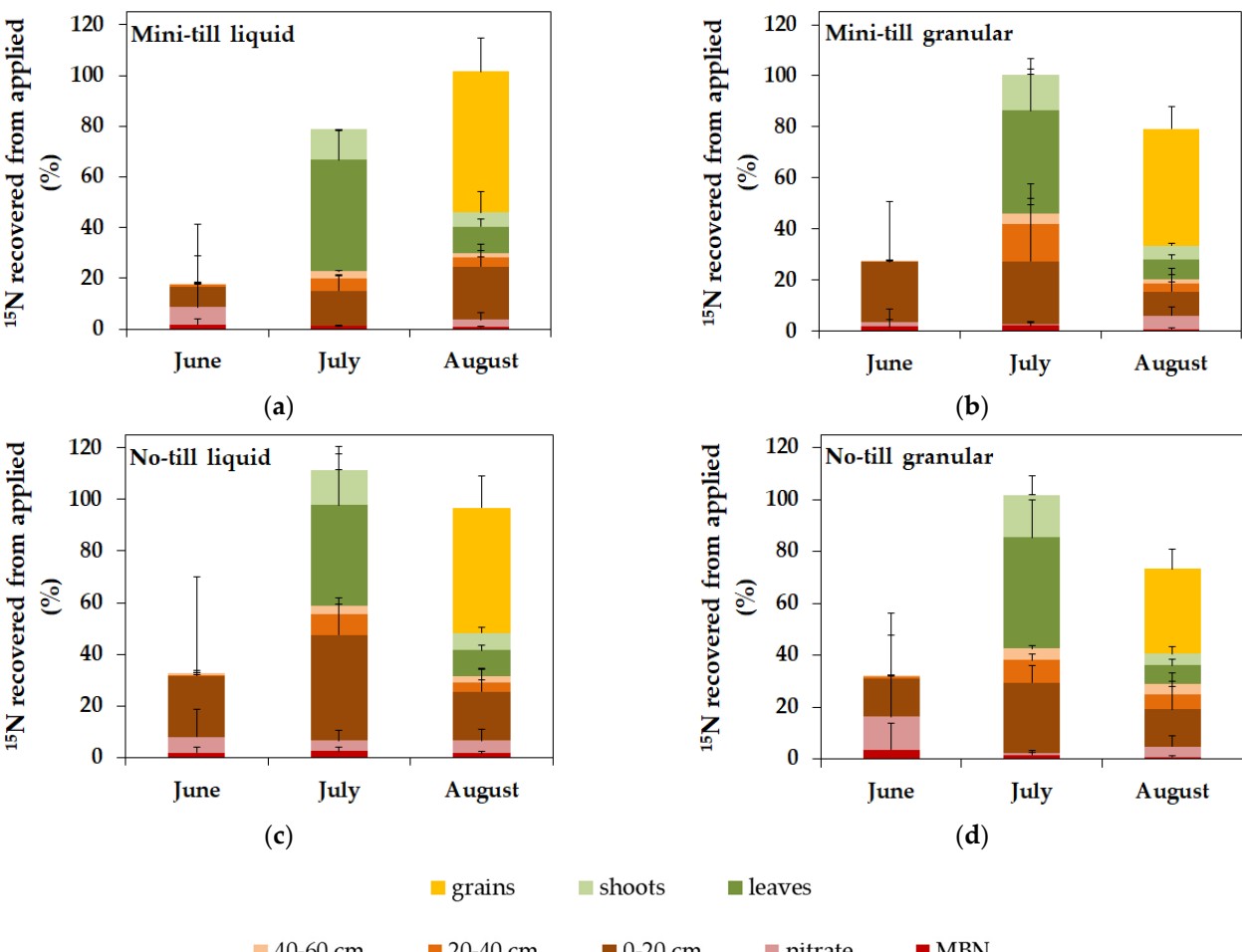

**Figure 2.** Recovery of applied $^{15}$N in soil and plant compartments over the vegetation period for the treatment combinations: (**a**) mini-till liquid, (**b**) mini-till granular, (**c**) no-till liquid, and (**d**) no-till granular. All data values are the means of n = 3 subplots. Error bars represent the standard deviation and are displayed on top of each bar to provide a better readability.

In mid July, 79% ± 14% to 112% ± 29% of the applied $^{15}$N was recovered (Figure 2). Of the 43% ± 17% of $^{15}$N found in the soil depth increments, 69% ± 9% was located at 0–20 cm (Figure 2). On average, 1% ± 2% and 2% ± 1% of the total recovered $^{15}$N was identified as nitrate and MBN, respectively. The fertilizer form used had no effect on the $^{15}$N recovery in the different soil depth increments. In contrast, the $^{15}$N recovery in 0–20 cm and nitrate was significantly higher under the no-till condition. Plants took up most of the applied fertilizer-derived $^{15}$N (56% ± 3%). Most of the total recovered $^{15}$N was found in the leaves (43% ± 13%). Fertilizer form had no effect on the plant $^{15}$N recovery.

In August, 73% ± 16% to 102% ± 15% of the applied $^{15}$N was recovered in total. Only 32% ± 8% $^{15}$N was left in the soil, of which 76% ± 7% was found at 0–20 cm. The $^{15}$N recovery in soil was not affected by the fertilizer form or by the tillage method, and translocation into deeper soil depth increments was similarly small for all treatments. In total, 6% ± 5% of the total recovered $^{15}$N was found as nitrate, whereas only 1% ± 1% $^{15}$N was immobilized as MBN. The fertilizer and tillage form had no significant effect on the recovery in nitrate or MBN fraction. In total, 68% ± 8% of the applied $^{15}$N was recovered in plants. Between July and August, a decreasing $^{15}$N abundance (of the total recovered $^{15}$N) in the leaves and shoots and increasing $^{15}$N abundance in the grains (52% ± 8%) mainly indicates N translocation occurring within the plant (Figure 2). The $^{15}$N recovery in grains

and in the whole plant was significantly higher under the mini-till and liquid fertilization conditions than under the no-till and granular fertilization conditions, respectively.

### 3.4. Plant–Microorganism Competition

Microbial biomass N-to-plant N ratios were around 2.2 (mean) in July and decreased to 1.3 in August (Figure 3), indicating a superior microbial N capture. The microbial biomass $^{15}$N-to-plant $^{15}$N ratios were much lower, at around 0.04 in July and 0.02 in August (Figure 3). Values below one indicate that developed plants increasingly exceeded microbial $^{15}$N immobilization. The lower ratios seen in August compared to July for both indexes suggest a stronger plant competitiveness. In July and August, both indexes were not affected by the fertilizer form used, but in August plant competitiveness was enhanced under the mini-till condition.

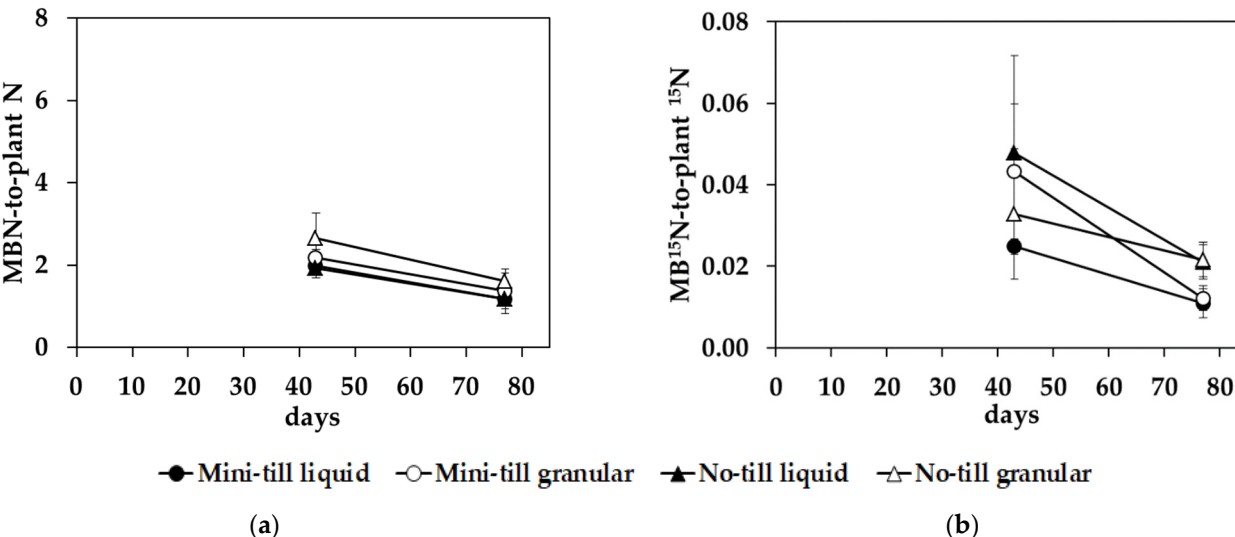

**Figure 3.** (**a**) Microbial biomass N-to-plant N ratio based on mg N m$^{-2}$ (**b**) and microbial biomass $^{15}$N-to-plant $^{15}$N ratio based on mg $^{15}$N m$^{-2}$. The mean of n = 3 subplots is shown; error bars represent the standard deviation.

## 4. Discussion

### 4.1. Initial Microbial Fertilizer-N Immobilization

Plant–microorganism competition for N depending on fertilizer form was investigated in clayey soils with a $^{15}$N labeling study over the vegetation period of spring wheat in the semi-arid climatic zone of North Kazakhstan. In our study, fertilizer-N was effectively kept in the topsoil, as reported for arable soils in semi-arid Australia and Canada [45,46]. The high total $^{15}$N recoveries in the plant and soil of 73% ± 16% to 102% ± 15% for the applied $^{15}$N in August show that the applied fertilizer-N was not lost due to (i) leaching, as precipitation with 92 mm (Figure 1) and hence translocation was small during the vegetation period (Figure 2), or (ii) volatilization, as NH$_3$ was applied as fertilizer in the soil, reducing NH$_3$ losses compared to near-surface applications [47].

Two days after fertilization, the MBN at 0–20 cm increased by 7 to 27% (Table 3) when the soil moisture was the largest (Supplementary Table S1), suggesting the occurrence of microbial N immobilization. The assumed microbial N immobilization is supported by findings from semi-arid Texas, where the addition of water and N has been shown to increase MBN shortly after application [48]. Microbial N immobilization can be confirmed by an increase in $^{15}$N excess in MBN after fertilization (Supplementary Table S2). Independently from fertilizer form, up to 4% of the applied $^{15}$N was immobilized by microorganisms (Figure 2), accounting for 8% of the total recovered fertilizer-N in June, excluding the top 4 cm, in which fertilizer was applied, as well as many microorganisms. Hence, the microbial fertilizer N immobilization was presumably even higher. Short-term competition studies in early vegetative plant stages report that microorganisms effectively took up most

fertilizer N, as shown in temperate soils in Inner Mongolia [44], in semi-arid prairie soils in the USA [16], and in semi-arid steppe soils in Inner Mongolia [17]. However, previous studies in more humid climate zones have shown that microorganisms immobilize up to 60% of applied fertilizer-N within three days [5,12,13]. The smaller microbial fertilizer-N immobilization at this early time point in our study was probably due to the excluded sampling of the top 4 cm, in which the fertilizer was applied. Two days after fertilization, the fertilizer was not yet well distributed throughout the soil, causing comparably smaller recoveries (Figure 2). However, 8% recovery in MBN from for the total recovered $^{15}$N in June show that if fertilizer-N became available to microorganisms, it was effectively immobilized.

In field and laboratory studies carried out under wetter conditions, it has been suggested that liquid N sources are more available to plants [18,20,49,50]. Hence, we assumed that in early vegetative stages liquid fertilizer-N in particular would be taken up in high amounts by microorganisms under the drier soil conditions of our study. In contrast to this, we could observe significantly higher MBN under granular fertilization in July, but not higher MBN or MB$^{15}$N in August (Table 3; Figure 2), when, in both cases, the soil moisture was low (Supplementary Table S1). Additionally, we did not observe significant differences in aboveground plant and grain yield (Table 4) when comparing the fertilizer forms, which disagrees with hypothesis one. This is probably because of the good water solubility of $NH_4NO_3$ [51], meaning that granular fertilizer had already been dissolved in the soil. Granular and liquid ammonium nitrate was hence similarly available to microorganisms and plants, independently of what form it was supplied in.

Interestingly, our study showed that the fertilizer form used was less important for the N uptake by microorganisms, only affecting plant and grain $^{15}$N recovery and NdfF in August according to ANOVA. The good solubility of granular ammonium nitrate in water (safety data sheet; C Roth, Karlsruhe, Germany) may have resulted in the similar performance of different forms of fertilizer. Observations showing a similar efficiency of liquid and granular fertilizer forms are scarce [52]. The higher plant recoveries of fertilizer-N and NdfF seen in August under mini-till management are in line with observations that under shallow tillage conditions more $^{15}$N was recovered in soil and plants than under zero tillage conditions for a urea fertilization rate of 100 kg N ha$^{-1}$ under a high precipitation level (178 to 232 mm) in the Canadian Great Plains [53]. In our study, this result can be attributed to the different plant development (Supplementary Figure S1a,b) occurring under no-till and mini-till management, where plants were further developed under the mini-till condition (Supplementary Figure S1a,b). The soil cracking (Supplementary Figure S1c) of these clayey soils (Table 1) in the dry summer 2019 was seen in abundance and occurred more often under the no-till condition, suggesting the occurrence of stronger soil compaction than under the mini-till condition. The higher initial bulk density seen under no-till management (Table 1) and increased compaction by soil drying could have increased the penetration resistance of the bulk soil (excluding dry cracks) and may therefore have resulted in smaller root development [54].

Consequently, we cannot accept our first hypothesis. Our results show that in these highly competitive conditions for N between plants and microorganisms with low levels of precipitation and high clay contents in North Kazakhstan, plants compete effectively against microorganisms for fertilizer-N (Figures 2 and 3) in the long run, regardless of the form in which N was initially supplied, though this is possibly affected by the tillage form used.

### 4.2. Plant–Microorganism Competition

In July, the MBN further increased (Table 3), but at this time point plants were already far developed (jointing stage) and took up high amounts of N (Table 3) regardless of the fertilizer form used. The MBN-to-plant N index (Figure 3a) shows that despite strong plant N uptake, microorganisms still hold 2.0 to 2.7 times more N than plants, indicating effective N immobilization by microorganisms. Four times higher values (about 10) for this

index were reported in a short-term competition study in July in a temperate grassland area in Inner Mongolia under nitrate addition [44]. Interestingly, the MB[15]N-to-plant [15]N ratio (Figure 3b) showed the opposite trend, with a stronger [15]N immobilization being seen in plants than in microorganisms. The MB[15]N-to-plant [15]N ratios fit ratios below 1 well, which had already been found three days after N application in a non-grazed semi-arid Inner Mongolian steppe soil in which the vegetation was already established [17], indicating the effective uptake of fertilizer-N by plants. However, three days after [15]N application, these ratios were higher (about 0.5) [17] than those seen in our study after 43 days. The higher [15]N recovery in plants and therefore smaller MB[15]N-to-plant [15]N ratios were due to the faster turnover times of microorganisms compared to plants [5,11]. Hence, the initially microbially immobilized [15]N was mineralized and released to the plant as suggested for tallgrass prairies [14]. The fact that plants outcompete microorganisms for N over longer time intervals could also suggest the existence of different N pools in plants and microorganisms, with plants preferring inorganic N while microorganisms predominantly take up organic N, as shown in various ecosystems [11,55]. In temperate grasslands, it has been shown that plants also compete effectively for a variety of amino acids [56,57]. However, inorganic N forms are still the major N source for plants in semi-arid regions [55], and incubation experiments suggest that microbial competition is more pronounced for organic than for inorganic N forms [58]. Consequently, competition for inorganic N could be avoided, as plants and microorganisms could prefer different sources of N. Despite N-recycling and the preferred N forms, the spatial aspect must also be considered. Fertilizer placement has been shown to strongly affect the uptake of N by plants [5,59]. The application of fertilizer into the furrow simultaneously with seeding could therefore have increased plant competition for fertilizer-N [59,60] compared to spatially more distributed microorganisms [15], meaning that microbial [15]N recovery is especially small at later time points due to this "dilution effect" of soil sampling.

From July to August, plants took up even more N (Table 3) and competed more effectively against microorganisms, as indicated by the higher [15]N recovery (Figure 2) and decreasing microorganism-to-plant N and [15]N ratios seen (Figure 3). Our study showed smaller competition indexes than were found in comparable studies [17,44]. This is probably due to the higher precipitation occurring in these studies (mean annual precipitation of 334 and 350 mm), as a higher water availability has been shown to increase MBN in semi-arid areas [48]. However, the differences in time periods between [15]N application and sampling time also have to be kept in mind. Whereas the [15]N competition index rapidly decreased from about 3 (24 h) to 0.5 (72 h) after [15]N application [17], we sampled after 43 and 77 days. Hence, plant–microorganism competition in our case was much more a product of numerous short-term competitions, and hence the competition ratios must have been smaller due to the death and remineralization of microorganisms. Unfortunately, no information about the clay content is offered in these studies [17,44], but the strong retention of mineral N seen in our study, especially abiotically of ammonium as another sink, might therefore have further increased the competition for the remaining N. In conclusion, in these clay-rich soils with very low levels of precipitation, plant–microorganism competition for N is further enhanced, meaning that our second hypothesis cannot be rejected.

**Supplementary Materials:** The following are available online at https://www.mdpi.com/article/10.3390/agronomy11122472/s1: Figure S1: Field observations. Table S1: Gravimetric water contents. Table S2: Absolute [15]N amount. Table S3: Results of ANOVA tests.

**Author Contributions:** Conceptualization, O.S. and M.K.; methodology, M.K., L.S.; software, M.K.; validation, O.S., J.F.C., C.F.S., and M.K.; formal analysis, M.K.; investigation, M.K., J.F.C., A.K., S.T.; resources, G.G., K.A., T.M.; data curation, O.S. and M.K.; writing—original draft preparation, M.K.; writing—review and editing, G.G., O.S., and M.K.; visualization, M.K.; supervision, O.S.; project administration, O.S.; funding acquisition, G.G. All authors have read and agreed to the published version of the manuscript.

**Funding:** This research was funded by the German Federal Ministry of Education and Research (BMBF), grant number FKZ 01LZ1704A.

**Institutional Review Board Statement:** Not applicable.

**Informed Consent Statement:** Not applicable.

**Data Availability Statement:** Data are available on request due to restrictions—e.g., privacy or ethical. The data presented in this study are available on request from the corresponding author. The data are not publicly available due to being part of an ongoing Ph.D. thesis.

**Acknowledgments:** We thank the team Soil Biology and Plan Nutrition of the University of Kassel for giving us the opportunity to conduct the chloroform fumigation extraction in their laboratory and for measuring these extracts. We are also grateful to Frank Schaarschmidt from the Institute of Biostatistics of the Leibniz University Hanover for helping us to develop the model used to account for spatial sampling. The publication of this article was funded by the Open Access Fund of the Leibniz Universität Hannover.

**Conflicts of Interest:** The authors declare no conflict of interest. T. Meinel is employed at Amazonen-Werke H. Dryer GmbH & Co. KG. The funders had no role in the design of the study; in the collection, analysis, or interpretation of data; in the writing of the manuscript; or in the decision to publish the results.

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
