# Peer review of "Competition of Plants and Microorganisms for Added Nitrogen in Different Fertilizer Forms in a Semi-Arid Climate"

_agronomy, doi:10.3390/agronomy11122472_

Round 1

Reviewer 1 Report

In my opinion the paper with title: “Competition of plants and microorganisms for added nitrogen 2 in different fertilizer forms in a semi-arid climate during the 3 vegetation period” will be better if the authors will make the corrections and specifications suggested in the review.

Line 126 - Please mention the concentration of NH415NO3 in the fertilizer solution in the liquid fertilizer variant.

In relation 3 for the calculation NdfF (%) the multiplication by 100 is to the numerator not to the denominator.

NdfF (%) = (15N at% excess in plant) *100 / (15N at% excess in fertilizer) (3)

Please make the correction in the relation.

In lines: 283, 288, 315 and others, it is mentioned that the percentage of nitrogen recovery is higher than 100%. Probably the exceedance is determined by the standard deviation 100 ± 2%. Please make the correction in the text.

283 - in “mid July, 79 to 112 % of the applied 15N was recovered (Figure 1)”.

288 – “In August, 73 to 102 % of the applied 15N were recovered in total…”, similar in line 315

Line 270 -The chapter name “3.3.15. N recovery “, is it correct?

Please add the caption for Figure 2 (a).

Please explain what δ15N represents (calculation relation), what is the significance of the parameter at% in relation 1. In relation 1 to the denominator there are too many closed parentheses.

Author Response

Please see the attatchment

Reviewer 2 Report

Dear authors,

Thank you for the opportunity to review this Manuscript (Competition of plants and microorganisms for added nitrogen in different fertilizer forms in a semi-arid climate during the vegetation period).  The authors tested four treatments (no-till granular and liquid, mini-till granular and liquid) to demonstrate the competition of plants and microorganisms in soil with added nitrogen. This study has great Figures to demonstrate the results, however, the authors did not use statistical analysis to explain the results. I am sending some suggestions to improve the study, and must be edited by the authors.

I wish you success in your research.

Introduction:

Explain the C:N in N immobilization by microorganisms

The authors could explore the no-till in the introduction.

The hypothesis was very clear that (i) liquid fertilizer can increase plant growth and N uptake in these semi-arid regions regardless of tillage form, thus increasing plant competition for N in the long run. Further, we assume that (ii) the plant-microorganism competition in semi-arid, clay-rich soils is more severe than reported for more humid regions where nutrients are better available.

The authors could explore the production of wheat in the crop rotation tested. Why the authors used this area?

Material and methods.

Explain and demonstrate in a table the area historic with crop rotation.

Explain the residue accumulation.

Please, calculate the C:N

Add, soil texture classification

This is not a great way to demonstrate data (30.05.2019). Please, edit it.

Demonstrate the precipitation during the experiment. Maybe, a Figure will be great.

Why the biomass N in soil was sampled in 0-20 cm depth? Please, check it. In the first paragraphs, the authors describe that “Soils of each subplot were sampled in triplicates directly under the furrow with cylindrical cores in 0-20, 20-40, and 40-60 cm (n = 9). For the 0-20 cm soil increment, the first 0-4 cm topsoil was not included in the soil sampling to avoid direct sampling of fertilizer.” This was not clear.

Explain better the soil collecting.

This is correct “the Tuckey’s honest”?

Result and discussion

The climate is the reason for “The highest but greatly deviating nitrate concentrations were observed in June after fertilizer application.” Please, add the precipitation data.

The authors should present the residue accumulation to explain the result? If there was no collecting, explain the plant development.

Explore N contents in the plant. Data is demonstrated in Table 1. The authors could run a statistical test to explain these N compartments.

What is “Total plant d.w.”

Table 3: The authors could run a statistical test to explain the results.

The data were not analyzed by a statistical test. This is not common in scientific studies. Because of that, there is no possibility to deliver information from this study.

The Results, Discussion, and Conclusion should be edited according to the statistical tests.

Author Response

Please see the attatchment.

Round 2

Reviewer 2 Report

The authors improved the manuscript.